Identification and transcriptomic profiling of salinity stress response genes in colored wheat mutant

Hong Min Jeong 1
Ko Chan Seop 1
Kim Jin-Baek 1
Kim Dae Yeon dykim@kongju.ac.kr 2
1 Advanced Radiation Technology Institute, Korea Atomic Energy Research Institute , Jeongeup , Jeollabuk-do , Korea
2 Plant Resources, Kongju National University , Yesan-eup , Chungnam , South Korea
Kutlu Imren
Electronic publication date: 2024 Mar 6
Publication date: 2024
Volume: 12
Electronic Location ID: e17043
Received 2023 Sep 22; Accepted 2024 Feb 13
Copyright: ©2024 Hong et al.
Copyright year: 2024
Copyright holder: Hong et al.
License: This is an open access article distributed under the terms of the Creative Commons Attribution License, which permits unrestricted use, distribution, reproduction and adaptation in any medium and for any purpose provided that it is properly attributed. For attribution, the original author(s), title, publication source (PeerJ) and either DOI or URL of the article must be cited.
License URL: https://creativecommons.org/licenses/by/4.0/

Keywords: Salinity stress, Gamma ray mutant, Salt-tolerant wheat, Transcriptome analysis, Gene regulation

Funding: The Korea Atomic Energy Research Institute Program 523420-24 Basic Science Research Program through the National Research Foundation of Korea (NRF) funded by the Ministry of Education 2022R1I1A1A01065420 This work was supported by the Korea Atomic Energy Research Institute Program (523420-24) and the Basic Science Research Program through the National Research Foundation of Korea (NRF) funded by the Ministry of Education (2022R1I1A1A01065420). The funders had no role in study design, data collection and analysis, decision to publish, or preparation of the manuscript.

==============================
Background

Salinity is a major abiotic stress that prevents normal plant growth and development, ultimately reducing crop productivity. This study investigated the effects of salinity stress on two wheat lines: PL1 (wild type) and PL6 (mutant line generated through gamma irradiation of PL1).

Results

The salinity treatment was carried out with a solution consisting of a total volume of 200 mL containing 150 mM NaCl. Salinity stress negatively impacted germination and plant growth in both lines, but PL6 exhibited higher tolerance. PL6 showed lower Na+ accumulation and higher K+ levels, indicating better ion homeostasis. Genome-wide transcriptomic analysis revealed distinct gene expression patterns between PL1 and PL6 under salt stress, resulting in notable phenotypic differences. Gene ontology analysis revealed positive correlations between salt stress and defense response, glutathione metabolism, peroxidase activity, and reactive oxygen species metabolic processes, highlighting the importance of antioxidant activities in salt tolerance. Additionally, hormone-related genes, transcription factors, and protein kinases showed differential expression, suggesting their roles in the differential salt stress response. Enrichment of pathways related to flavonoid biosynthesis and secondary metabolite biosynthesis in PL6 may contribute to its enhanced antioxidant activities. Furthermore, differentially expressed genes associated with the circadian clock system, cytoskeleton organization, and cell wall organization shed light on the plant’s response to salt stress.

Conclusions

Understanding these mechanisms is crucial for developing stress-tolerant crop varieties, improving agricultural practices, and breeding salt-resistant crops to enhance global food production and address food security challenges.

Introduction

Wheat is a crucial crop cultivated globally, contributing to 30% of global grain production and providing approximately 20% of the calories consumed by humans (Shiferaw et al., 2013; Seleiman et al., 2022). Soil salinity poses a critical issue, resulting in yield losses of up to 60% in wheat production (El-Hendawy et al., 2017). The impact of salinity is extensive, with >20% of irrigated land worldwide being affected (EL Sabagh et al., 2021). Furthermore, it is projected that up to 50% of arable land will be lost by 2,050 owing to salinization caused by both human activities and ongoing climate change (Asif et al., 2018; Kumar & Sharma, 2020; Chele et al., 2021).

Salinity stress disrupts plant growth by increasing Na+ ion assimilation and reducing the Na+/K+ ratio, leading to osmotic stress and ion toxicity, consequently affecting normal plant development (EL Sabagh et al., 2021). Additionally, under salinity stress, oxidative stress can impair plant growth through reduced photosynthetic capacity, oxidative damage caused by an imbalance in reactive oxygen species (ROS) production, and decreased antioxidant activity, ultimately leading to reduced crop yield (Hasanuzzaman et al., 2014; Sadak, 2019; Omrani et al., 2022).

Numerous studies have focused on breeding new salt-tolerant crop varieties using molecular and biotechnologiesand on selecting salt-tolerant crops (Huang et al., 2008; Ismail & Horie, 2017; Saade et al., 2020; Hussain et al., 2021). These selection criteria encompass germination rate, plant growth, chlorophyll content, and K+/Na+ ratio (El-Hendawy et al., 2019; Choudhary et al., 2021; Tsai et al., 2019; Assaha et al., 2017; Singh & Sarkar, 2014). Particularly, the germination and growth rates during the early stages of plant development have proven useful for screening salt-tolerant crops (Choudhary et al., 2021). In Brassica napus, root and shoot lengths act as early indicators for evaluating salt tolerance (Long, Zou & Zhang, 2015). In rice, salt-tolerant cultivars have higher chlorophyll content and Na+/K+ ratios under salt stress conditions than salt-susceptible cultivars (Singh & Sarkar, 2014). Regulating excessive Na+ accumulation in plants is a vital strategy for enhancing salt resistance (Tester & Davenport, 2003; Møller & Tester, 2007; Møller et al., 2009). The high-affinity K+ transporter (HKT) gene family plays a crucial role in maintaining Na+ and K+ balance in plant growth, development, abiotic stress responses, and salt tolerance (Horie, Hause & Schroeder, 2009; Li et al., 2019; Riedelsberger et al., 2021). Initially identified in wheat (Schachtman & Schroeder, 1994), HKT genes have been found to reduce Na+ accumulation in higher plants, such as Arabidopsis, rice, and wheat (Riedelsberger et al., 2021). Additionally, the salt overly sensitive (SOS) gene family is involved in regulating ion homeostasis and Na+ exclusion at the cellular level, affecting plant salinity tolerance (Yang et al., 2009).

Despite ongoing research on gene regulation under salt stress, limited progress has been made in establishing appropriate screening methods using genetic resources, understanding mechanisms underlying osmotic stress/tissue resistance, and identifying salt-tolerant crops (Genc et al., 2019). Furthermore, as elite germplasm may lack genes that confer salt resistance, genetic engineering involving the artificial insertion of specific genes may be required to develop new crop varieties (Colmer, Flowers & Munns, 2006; Shavrukov, Langridge & Tester, 2009; Munns et al., 2012; Deinlein et al., 2014).

Genetic diversity is crucial for developing new and improved crop varieties with desirable traits. However, breeders often focus on improving traits by selecting offspring with the best attributes, leading to a decrease in genetic diversity when some plants become vulnerable to environmental stresses. Mutation breeding is a widely used method for enhancing genetic diversity and improving crop traits. Gamma rays, being physical mutagens, are commonly used for plant mutation breeding and have been instrumental in developing >50% of the 3,401 new varieties included in the FAO/IAEA Mutant Variety Database (https://nucleus.iaea.org/sites/mvd/SitePages/Home.aspx). In light of these findings, the construction of a mutant pool using gamma rays offers an opportunity to develop salt-resistant wheat by securing genetic diversity. Colored wheat possesses advantages such as enhanced nutritional content, increased antioxidant levels, and potential health benefits due to its rich phytochemical profile. Colored wheat possesses advantages such as enhanced nutritional content, increased antioxidant levels, and potential health benefits due to its rich phytochemical profile (Hong et al., 2019; Garg et al., 2022). Investigating the salt tolerance in colored wheat, which is characterized by these valuable features, and revealing the distinctive features that contribute to its resistance to salinity are essential to obtain new genetic resources with high salt tolerance and nutrition. This study aims to investigate the changes in the salt tolerance mechanism in the colored wheat line we developed through gamma-ray mutation.

Materials & methods

Plant materials

One of the progenies resulting from the cross between ‘Woori-mil’ (obtained from the National Agrobiodiversity Center, RDA, Korea; accession no. IT172221) and ‘D-7’ (an inbred line developed by Korea University; Fleming4/3/PIO2580//T831032/Hamlet) exhibited color segregation. This specific progeny, carefully chosen, had spikes containing colored seeds, and these were utilized in the subsequent generation. Finally, we developed the common wheat (Triticum aestivum L., 2n = 6x = 42, AABBDD) inbred line K4191 (hereafter termed PL1), distinguished by its deep purple grain color. K4191 was derived from the F4:8 generation resulting from the cross between ‘Woori-mil’ and ‘D-7’, both of which have common seed color (Hong et al., 2019). To induce genetic variation and diversity the population of colored wheat, colored wheat seeds (PL1) were irradiated with 200 Gy gamma rays at a dose rate of 25 Gy/h using a 60Co gamma irradiator (150 TBq of capacity; Noridon, Ottawa, ON, Canada) at the Korea Atomic Energy Research Institute. Subsequently, the irradiated seeds were planted at the radiation breeding research farm. Briefly, 1500 M0 seeds were exposed to irradiation, and the resulting seeds were sown to generate the M1 generation. Among these seeds, 287 phenotypically distinctive lines were carefully selected with one spike per plant, and mutation breeding spanning from M0 to M4 was performed as thoroughly described in a previous study (Hong et al., 2019). The resulting mutants were continuously cultivated up to the M6 generation and carefully selected based on excellent agricultural traits, including flowering time, plant height, yield, and grain color. In total, 50 mutant lines displaying stable phenotypes for at least two generations were chosen for further salt-tolerance screening.

Salt stress treatment

The PL1 (control line, K4191) and PL6 (mutant line) seeds were surface-sterilized with 70% ethanol for 1 min and then washed with sterile distilled water. Subsequently, the seeds were placed on moist filter papers in a Petri dish (SPL Life Sciences) until the first leaf of the seedlings appeared. Next, the uniformly germinated seeds were transferred to Incu Tissue culture vessels (SPL Life Sciences) filled with half-strength Hoagland’s culture solution (Sigma-Aldrich, St. Louis, MO, USA). The solutions were replaced daily. The seedlings were grown for 7 days in a well-controlled chamber at 22 °C and 60% humidity, with a photoperiod regime of 16/8 h day/night at 200–300 µmol m−2s−1 light. After 7 days of transplanting, the seedlings were subjected to a salt stress treatment of a total volume of 200 mL of the solution containing 150 mM NaCl. Following treatment with 150 mM NaCl, the wheat leaves were collected at 0, 3, 24, and 48 h. Both control and salt-stressed seedlings were collected individually. The samples were immediately frozen in liquid nitrogen and stored at −80 °C until use in further experiments.

Measurement of leaf Na+ and K+ contents

The wheat leaves were collected separately and immediately frozen in liquid nitrogen. Subsequently, the samples were freeze-dried for 3 days in a Freeze Dry System (IlshinBioBase, Dongducheonsi, Gyeongi, Korea). The freeze-dried samples were then finely ground into a powder using a mortar and pestle. For further analysis, 50 mg of the freeze-dried samples was weighed using an analytical balance and boiled for 2 h at 200 °C in three mL of HNO3 (70%, v/v) for digestion. After digestion, the extracted samples were diluted with 5% HNO3 and filtered through a hydrophilic polytetrafluoroethylene syringe filter (0.45-μM pore size, 25-mm diameter). The shoot Na+ and K+ contents were measured using inductively coupled plasma atomic emission spectroscopy (ICP-AES, 720 series; Agilent, Santa Clara, CA, USA) and quantitatively analyzed using a VistaChip CCD detector (Agilent).

Measurement of chlorophyll content

To determine the chlorophyll content, wheat seedling samples were extracted with 100% methanol at 4 °C. The sample extracts were then subjected to centrifugation at 12,000 × g for 10 min, and the supernatant was used for chlorophyll content analysis. The total chlorophyll, chlorophyll a, and chlorophyll b concentrations were determined by measuring the absorbance at 644.8 and 661.6 nm using a UV–VIS spectrophotometer (Lichtenthaler, 1987). The chlorophyll concentration was calculated using the following equations: (1) Ca=11.24×A661.6−2.04×A644.8

(2) Cb=20.13×A644.8−4.19×A661.6

(3) Ctotal=7.05×A661.6+18.09×A644.8

where Ca, Cb, and Ctotal denote the concentrations of chlorophyll a, chlorophyll b, and total chlorophyll, respectively.

RNA sequencing and gene expression analyses

Total RNA was extracted from the wheat leaves of both PL1 and PL6 at each timepoint (0, 3, 24, and 48 h) using TRIzol (Invitrogen, Carlsbad, CA, USA) following the manufacturer’s instructions. Two independent biological replicates were performed for each timepoint and line to ensure the reliability and reproducibility of the RNA-seq data. Additionally, the extracted RNA samples were treated with DNase I to remove any potential genomic DNA contamination. The RNA quality was assessed using an Agilent 2100 bioanalyzer (Agilent Technologies, Amstelveen, The Netherlands), and RNA quantification was performed using an ND-2000 Spectrophotometer (Thermo Inc.; Wilmington, DE, USA). For constructing the RNA-seq paired-end libraries, 10 µg of total RNA extracted from the samples was used with the TruSeq RNA Sample Preparation Kit (Catalog #RS-122-2001; Illumina, San Diego, CA, USA). The mRNA was isolated using a Poly(A) RNA Selection Kit (LEXOGEN, Inc.; Vienna, Austria) and reverse-transcribed into cDNA following the manufacturer’s instructions. The libraries were assessed using the Agilent 2100 bioanalyzer, and the mean fragment size was evaluated using a DNA High Sensitivity Kit (Agilent, Santa Clara, CA, USA). High-throughput sequencing was conducted using the HiSeq 2000 platform (Illumina). Before alignment, adaptor sequences were removed, and sequence quality was evaluated using the Bbduk tool (minimum length >20 and Q >20; https://jgi.doe.gov/data-and-tools/software-tools/bbtools/bb-tools-user-guide/bbduk-guide/). The reads were aligned to the wheat genome sequence provided by the International Wheat Genome Sequencing Consortium (IWGSC) wheat reference sequence (IWGSC Reference Sequence v1.0; https://urgi.versailles.inra.fr/download/iwgsc/IWGSC_RefSeq_Annotations/v1.0/) using the HISAT2 alignment program with default parameters (Kim, Langmead & Salzberg, 2015). Reads mapped to the exons of each gene were enumerated using the HTSeq v0.6.1 high-throughput sequencing framework (Anders, Pyl & Huber, 2015). Subsequently, the differentially expressed genes (DEGs) under salt stress and control conditions were identified using the EdgeR package (Robinson, McCarthy & Smyth, 2010). Upregulated and downregulated genes with a p-value of <0.05, false discovery rate (FDR) of <0.05, and an absolute fold change value of >2 were used for downstream functional analysis. The log2-transformed transcript per million values were calculated using TPMCalculator (Vera Alvarez et al., 2019), and heatmaps of DEGs under control and stress conditions were generated. Local BlastX was used with peptide sequences of the Poaceae family retrieved from the National Center for Biotechnology Information (NCBI) database using an e-value threshold of 1 × 10−5 to annotate the DEGs. For gene expression analysis, total RNA was used to synthesize first-strand cDNA using the Power cDNA Synthesis Kit (iNtRON Biotechnology, Gyeonggi-do, Korea). Reverse transcription-quantitative polymerase chain reaction (RT-qPCR) was performed in a total volume of 20 μL containing 1 μL of cDNA template, 0.2 μM primers, and 10 μL of TB Green Premix Ex Taq II (Takara, Kusatsu, Shiga, Japan). RT-qPCR was conducted using a CFX96TM Real-time PCR system (Bio-Rad, Hercules, CA, USA) with the following program: 95 ° C for 5 min, followed by 40 cycles at 95 °C for 10 s and 65 °C for 30 s. Actin (AB181991) was used as an internal control. The primers used in this experiment are listed in Table S1.

Functional analysis of DEGs

All expressed genes under both control and stress conditions were subjected to Gene Set Enrichment Analysis (GSEA) using the GSEA software (Subramanian et al., 2005). The gene matrix transposed file format (.GMT) of wheat was downloaded from g:Profiler (https://biit.cs.ut.ee/gprofiler/gost), a web server for functional enrichment analysis and gene list conversion (Raudvere et al., 2019). The enrichment score of each gene set was calculated using the full ranking, and the normalized enrichment score (NES) was determined for each gene set. The GSEA results, including rank, expression, and class files, were visualized as a network using Enrichment Map (Merico et al., 2010). For Kyoto Encyclopedia of Genes and Genome (KEGG) pathway enrichment analysis, the KEGG Orthology Database in KOBAS-i was used to predict the putative pathways of DEGs (Bu et al., 2021). The plant transcription factor data were obtained from the Plant Transcription Factor Database (PlantTFDB) (Tian et al., 2020). Protein Basic Local Alignment Search Tool (BLASTP) was used on the peptide sequences of the DEGs, based on the local transcription factor database obtained from PlantTFDB, with an E-value threshold of 1 × 10−1 and sequence identity of >80%. Mev software (http://sourceforge.net/projects/mev-tm4/files/mev-tm4/) was used for k-means clustering of DEGs identified from the GSEA, KEGG pathway, and transcription factor analyses. The results of the GSEA and KEGG pathway analysis were generated using an R script and the ggplot2 R package. Additionally, MapMan was used to identify the pathways of stage-specific genes (Sreenivasulu et al., 2008).

Enzyme activities assays

The crude enzyme was extracted from 100 mg of wheat leaves using a protein extraction buffer containing 50 mM potassium phosphate buffer (pH 7.5). The activities of catalase (CAT), peroxidase (POD), and superoxide dismutase (SOD) and total antioxidant activity (TAC) were measured using commercially available assay kits. Specifically, CAT activity was determined using a catalase microplate assay kit (kit number: MBS8243260; MyBiosource, Inc., San Diego, CA, USA), POD activity was measured using a POD assay kit (kit number: KTB1150; Abbkine, Inc., Wuhan, China), and SOD activity was estimated using a total SOD activity assay kit (WST-1 method) (kit number: MBS2540402; MyBiosource, Inc., San Diego, CA, USA). TAC was assessed using a TAC assay kit (kit number: MAK187; Sigma-Aldrich, St. Louis, MO, USA). The preparation of the reaction mixture and the calculations for each measurement were performed as described in the respective protocol books provided with each assay kit.

Results

Characteristics of the salt-tolerant colored wheat mutant induced via gamma irradiation

Throughout the mutation breeding process, detailed records of agricultural traits, including the flowering time, plant height, and yield, were meticulously collected for the mutant lines. These data allowed for a comprehensive assessment of the phenotypic characteristics of the lines. Evidence supporting the stable phenotype of the mutant lines is provided in Fig. S1, which also presents the field performance of PL1 and PL6. Additionally, the difference in grain color between the colored wheat lines used in this study is illustrated in Fig. S2. To select salt-resistant wheat lines, 50 wheat mutant lines were treated with 150 mM NaCl, and germination rate, shoot length, and root length were subsequently investigated. The results showed that 27 lines exhibited a germination rate of 90%. Among these, six lines demonstrated a growth increase of approximately 20% or more compared to the control group subjected to salt treatment (Fig. S3). Through preliminary salt-tolerance screening, PL6 was selected as the gamma ray-derived mutant line that exhibited favorable salt-tolerance characteristics (Fig. S3). To assess the growth response of the control line (PL1) and PL6 under varying salt concentrations, the seeds were treated with NaCl solutions of 50, 100, 150, 200, 250, 300, and 500 mM, along with distilled water as the control (Choudhary et al., 2021). Overall, high salt concentrations negatively affected seed germination and seedling growth (Figs. 1A and 1B). The germination percentage and seedling growth were reduced with increasing salt concentration in both PL1 and PL6 (Table S2). However, PL6 demonstrated higher germination percentages, particularly at the maximum NaCl concentration, exceeding those of PL1. Remarkably, a maximum increase of 20% in germination was observed for PL6 following treatment with 250 mM NaCl. Moreover, PL6 consistently outperformed PL1 in terms of seedling growth under all salt treatment conditions, as evidenced by its longer shoot and root lengths (Figs. 1C and 1D). The comprehensive data strongly indicates that the gamma ray-derived mutant PL6 exhibits higher resistance to salt stress than PL1.

Figure 1 Effect of salt stress on seed germination and seedling growth.

(A) Germination rate of wheat seeds under different salt concentrations. 500 seeds from each line were placed on two layers of germination paper and exposed to a solution containing 150 mM NaCl in a phytohealth chamber (SPL Life Sciences) at a temperature of 22° C. Germination was assessed after 4 days. (B) Wheat seedling growth under different salinity levels. Seven-day-old seedlings were subjected to a salt stress treatment with a total volume of 200 ml of the solution containing 150 mM NaCl after 4 days. (C) Phenotypes of wheat seedlings under different salinity levels after 4 days of salt stress with a total volume of 200 ml of the solution containing 150 mM NaCl. (D) Shoot and root lengths of wheat seedlings under different salinity conditions after 4 days of salt stress with a total volume of 200 ml of the solution containing 150 mM NaCl. Independent t-tests demonstrated significant differences (*p < 0.05 and **p < 0.01).

Assessment of Na+, K+, and chlorophyll contents under salt stress conditions

Prior to treatment, PL6 had a higher Na+ ion content than PL1 (Fig. 2A). However, with increasing time of exposure to salt stress, the Na+ ion content markedly increased in both PL1 and PL6. Notably, the rate of increase in Na+ ion content was lower in PL6 than in PL1. Conversely, the K+ ion content steadily decreased with salt treatment in both PL1 and PL6 (Fig. 2B). To further analyze the ion contents, we calculated the relative ratios of K+ and Na+ ions in PL1 and PL6, considering their respective contents under control conditions (Figs. 2C and 2D). In PL1, the Na+ ion content increased significantly by 47 times from the baseline (0 h) to 48 h following salt treatment. In contrast, PL6 exhibited a milder increase in Na+ ion content, approximately 20 times higher at 48 h after salt stress. Consequently, the relative Na+ content was more profoundly affected by salt stress in PL1 than in PL6. Interestingly, the chlorophyll concentrations of both PL1 and PL6 remained relatively stable under salt stress (Fig. 2E), indicating that they were not significantly affected by the imposed salinity conditions.

Figure 2 Na+ and K+ ion contents, differential ratios of K+ and Na+ and chlorophyll concentrations for PL1 and PL6 under salt stress treatment.

Seven-day-old seedlings were subjected to a salt stress treatment with a total volume of 200 ml of the solution containing 150 mM NaCl. After treatment with 150 mM NaCl, wheat leaves were collected at 3, 24, and 48 h. (A) Na+ ion content in the shoots under different salt stress exposure times. (B) K+ ion content in the shoots under different salt stress exposure times. (C) Changes in the relative “Na+ ratio” in shoots at different time points after salt stress treatment. The “Na+ ratio” represents the relative proportion of Na+ content in shoots compared to the Na+ content at 0 h (baseline). Data points at 3 h, 24 h, and 48 h indicate the fold change of Na+ content in shoots compared to the baseline (0 h). (D) Changes in the relative “K+ ratio” in shoots at different time points after salt stress treatment. The “K+ ratio” represents the relative proportion of K+ content in shoots compared to the K+ content at 0 h (baseline). Data points at 3 h, 24 h, and 48 h indicate the fold change of K+ content in shoots compared to the baseline (0 h). (E) Chlorophyll concentrations in the shoots under different salt stress exposure times. Each bar represents the mean ±standard error (n = 3). Independent t-tests showed significant differences (*p < 0.05 and **p < 0.01).

DEGs during salt stress

After treatment with 150 mM NaCl, leaves were harvested from PL1 and PL6 at 0, 3, 24, and 48 h and subjected to RNA sequencing. Following quality evaluation and trimming, an average of 38.1 million trimmed reads and over 22.1 billion bases were generated from each sample under both control and salt stress conditions. The average percentage of Q20 and Q30 bases was found to be 98.4% and 95.5%, respectively, indicating high sequencing quality. Moreover, >96% of the sequenced data exhibited an average mapping rate of 96.16%, successfully aligning to the IWGSC wheat reference sequence (Table S3). During data analysis, a total of 4,017 DEGs were identified with a p-value of <0.05, FDR of <0.05, and absolute fold change value of >2 (Fig. 3A and Table S4). Specifically, in PL1, 872, 1,588, and 1,080 DEGs were identified at 3, 24, and 48 h after salt treatment, respectively, in comparison to the untreated condition (0h) (Fig. 3B). Similarly, for PL6, the number of DEGs was 566, 1,248, and 1,810 at 3, 24, and 48 h after salt treatment, respectively (Fig. 3C). These results highlight the dynamic gene expression changes in PL1 and PL6 under salt stress at different timepoints, contributing to a better understanding of the underlying molecular responses to salt stress in these wheat lines.

Figure 3 Differentially expressed genes (DEGs) and Gene Set Enrichment Analysis (GSEA) for PL1 and PL6.

(A) Venn diagrams showing the number of DEGs between PL1 and PL6 and the overlap of all DEGs at different time points after exposure to salt stress. (B) Number of DEGs only expressed in PL1 at different time points after exposure to salt stress. (C) Number of DEGs only expressed in PL6 at different time points after exposure to salt stress. (D) GSEA enrichment analysis with gene ontology of the DEGs. Dots indicate significant GO terms from the pairwise gene set enrichment analysis comparisons at each time point after exposure to salt stress. The size of the dots indicates the number of genes, and the color of the dots indicates the −log10 FDR value within the pathway.

Functional analysis of the DEGs during salt stress

Overall, 33 GO terms were identified for each treatment condition (Fig. 3D and Table S5). Notably, several gene sets, including defense response (GO: 0006952), glutathione metabolic process (GO: 0006749), peroxidase activity (GO: 0004601), ROS metabolic process (GO: 0072593), response to biotic stimulus (GO: 0009607), and response to stress (GO: 0006950), were positively correlated with salt stress and PL6, exhibiting a positive NES (Fig. 3D). To visualize the results, all the gene sets from the GSEA were organized into four networks using Enrichment Map (Merico et al., 2010) (Figs. 4A–4E). The expression patterns of each network in Figs 4A–4E for PL1 and PL6 were clustered by expressed patterns (Figs. 4F–4J, respectively). The K-means clustering algorithm in the Mev software was used to identify the clusters of DEGs in each GO term under control and salt stress conditions based on their expression patterns. Most of the expression patterns from the identified clusters did not differ between the control and salt stress conditions. Three clusters that demonstrated different expression patterns for PL1 and PL6, especially those upregulated in PL6, were selected and marked in red boxes in Figs. 4F, 4G, and 4I, and a heatmap of the genes from these clusters was generated (Fig. 4K). Plant hormone-related genes (TRAESCS1B02G145800 and TRAESCS1B02G138100), ROS-related genes (TRAESCS1B02G059100, TRAESCS1B02G095800, TRAESCS1B02G096200, TRAESCS1B02G096900, and TRAESCS1B02G115900), and stress-response genes (TRAESCS5D02G492900, TRAESCS1A02G009900, TRAESCS1B02G023000, and TRAESCS2A02G037400) were highly expressed in PL6 under salt stress conditions. Furthermore, six genes related to chromatin remodeling (TRAESCS1B02G048900, TRAESCS1B02G049100, TRAESCS1D02G286700, TRAESCS1B02G149000, and TRAESCS7D02G246600) showed high expression patterns in PL6 under salt stress conditions [Table 1]. A high number of transcriptomes of MADS-box transcription factors (TRAESCS4A02G002600, and TRAESCS6D02G293200) were also detected in PL6 under salt stress. An auxin-responsive protein (TRAESCS1B02G138100) and probable histone H2A variant 3 (TRAESCS7D02G246600) were also found in cluster 4 [Table 1].

Figure 4 Gene ontology (GO) enrichment map and differential gene expression profiling for PL1 and PL6.

(A–E) Five networks of significantly enriched gene sets between PL1 and PL6 obtained on the enrichment map. Nodes representing enriched gene sets were classified based on their similarity to related gene sets. The size of the node is proportional to the total number of genes. The thickness of the green line between nodes represents the proportion of shared genes between gene sets. (F–J) The expression patterns of each network at each time point after exposure to salt stress. Each cluster represents a group of functionally related gene sets that showed similar expression patterns. Figures 4F, 4G, 4H, 4I, and 4J show multiple clusters derived from the networks of Figs. 4A, 4B,4C, 4D, and 4E, respectively. Clusters showing different expression patterns between PL1 and PL6 were indicated in red boxes. (K) Heatmaps representing the expressions of differentially expressed genes (DEGs) marked in red boxes (F, G, and I) for PL1 and PL6.

In the case of the differences in the KEGG pathways between PL1 and PL6 under salt stress conditions, the rich factor of “Biosynthesis of secondary metabolites” in PL6 after 3 h of salt stress was ∼0.05, increasing to ∼0.2 after 48 h of salt stress (Fig. 5). Likewise, the rich factors of “Flavonoid biosynthesis” were 0.17 and 0.23, after 24 and 48 h of salt stress, respectively. This was only observed in PL6 during salt stress conditions (Fig. 5 and Table S6).

Table 1 List of differentially expressed genes selected by K-means clustering from GSEA analysis.

Gene ID	Description	Length	E-value	Similarity (%)	Log2 fold change (PL6/PL1)	p-value	FDR	Group	
					0 h	3 h	12 h	24 h				
TRAESCS1B02G145800	abscisic acid receptor PYL8	205	1E−149	92.57	4.37	1.96	7.28	1.87	2.09E−19	2.53E−29	DEGs
in red boxes
of cluster 4
in Fig. 4F	
TRAESCS1B02G138100	auxin-responsive protein IAA15	198	2E−143	81.71	3.64	1.88	3.57	2.42	1.56E−28	6.22E−36	
TRAESCS5D02G129700	chaperone protein dnaJ GFA2, mitochondrial	421	0	84.11	8.75	1.57	6.56	5.69	4.77E−38	7.45E−69	
TRAESCS1B02G018100	defensin	81	1.1E−38	84.72	4.86	7.74	5.55	−0.43	1.82E−09	4.81E−93	
TRAESCS1B02G059100	dehydroascorbate reductase	212	9E−156	95.77	3.75	3.64	3.78	1.76	4.05E−69	4.52E−24	
TRAESCS1B02G050200	E3 ubiquitin-protein ligase XB3	486	0	82.96	3.96	3.92	4.48	1.15	1.98E−36	9.28E−17	
TRAESCS5D02G492900	heat shock cognate 70 kDa protein 2	614	0	92.5	3.67	1.74	3.52	3.68	6.04E−97	2.97E−66	
TRAESCS1B02G037100	NAD(P)-binding Rossmann-fold superfamily protein	300	0	91.03	5.36	4.72	5.49	2.27	5.67E−67	1.12E−26	
TRAESCS1B02G095800	Peroxidase 2	340	0	89.71	7.85	1.71	7.13	2.85	8.69E−72	9.78E−22	
TRAESCS1B02G096200	peroxidase 5	338	0	85.36	4.99	1.58	4.80	1.88	1.81E−23	3.68E−64	
TRAESCS1B02G096900	Peroxidase 5	343	0	75.36	4.34	1.12	3.82	3.15	7.29E−26	8.91E−18	
TRAESCS1B02G115900	peroxidase A2-like	342	0	82.95	3.84	2.45	4.16	3.27	5.42E−59	3.96E−69	
TRAESCS1B02G038700	protein NRT1/ PTR FAMILY 6.2	582	0	90.52	7.36	8.20	8.09	1.46	1.78E−64	2.32E−34	
TRAESCS1A02G009900	putative disease resistance RPP13-like protein 3	844	0	82.63	6.88	6.49	6.40	−1.81	1.3E−14	2.22E−56	
TRAESCS1B02G023000	putative disease resistance RPP13-like protein 3	920	0	72.15	7.98	8.61	7.67	2.83	2.09E−29	9.97E−62	
TRAESCS1B02G102200	replication protein A 70 kDa DNA-binding
subunit C-like	881	0	73.55	4.59	0.95	7.35	2.00	2.3E−18	1.6E−27	
TRAESCS2A02G037400	stress-response A/B barrel domain-containing protein HS1	115	9E−76	85.48	8.11	1.65	8.58	8.10	2.94E−31	2.53E−29	
TRAESCS1B02G034100	subtilisin-chymotrypsin inhibitor CI-1B	74	1.8E−44	84.47	9.97	11.00	11.50	3.89	4.45E−72	3.44E−77	
TRAESCS1B02G035100	subtilisin-chymotrypsin inhibitor CI-1B	74	2.3E−45	84.72	10.70	11.55	12.07	2.56	2.35E−80	4.03E−13	
TRAESCS4D02G170100	60S ribosomal protein L19-1	228	2E−144	88.22	7.03	−0.34	7.73	−7.05	5.65E−47	1.2E−44	DEGs
in red boxes
of cluster 3
in Fig. 4G	
TRAESCS2A02G027000	actin-related protein 9 isoform X1	526	0	84.59	5.48	1.33	3.59	2.19	6.68E−81	1.05E−77	
TRAESCS1B02G133100	DNA-directed RNA polymerases II, IV and V subunit 11	119	2E−86	96.77	8.01	2.08	8.31	2.82	7.37E−23	3.82E−21	
TRAESCS2D02G596000	exocyst complex component EXO70A1	637	0	93.63	6.65	2.49	5.61	2.78	7.27E−83	1.44E−79	
TRAESCS1B02G048900	histone H2A	154	1E−99	92.72	5.66	5.03	6.05	3.09	2.91E−11	6.97E−10	
TRAESCS1B02G049100	histone H2A	155	1E−98	91.26	7.65	8.17	8.49	3.42	4.73E−34	4.76E−32	
TRAESCS1D02G286700	histone H4	103	6.6E−70	99.76	3.41	−1.75	2.65	0.70	6.57E−35	6.97E−33	
TRAESCS1B02G149000	INO80 complex subunit D	288	0	82.89	7.67	1.69	7.82	2.84	6.2E−18	3.82E−21	
TRAESCS1B02G126900	NAD(P)H-quinone oxidoreductase subunit S, chloroplastic	239	6E-167	82.95	3.54	1.97	3.25	1.86	3.85E−37	2.43E−16	
TRAESCS2A02G046200	nuclear transport factor 2 (NTF2)-like protein	199	4E−115	82.59	8.80	2.01	8.64	8.71	6.74E−30	6.97E−33	
TRAESCS1A02G403800	predicted protein	266	0	96.99	2.85	2.44	2.78	1.30	0.006931	1.05E−77	
TRAESCS7D02G370400	predicted protein	312	0	83.35	2.11	−0.21	3.22	2.03	8.65E−36	5.3E−28	
TRAESCS7D02G246600	probable histone H2A variant 3	139	3E−95	95.7	2.08	1.78	3.65	2.75	1.62E−42	1.44E−79	
TRAESCS3A02G516500	Protein COFACTOR ASSEMBLY OF COMPLEX C SUBUNIT B CCB3, chloroplastic	192	4E−117	76.6	7.91	2.26	8.02	8.40	4.51E−69	3.25E−66	
TRAESCS 1B02G100800	RNA-binding protein 8A	209	9E−152	85.74	4.77	1.01	4.64	2.08	7.57E−28	1.2E−44	
TRAESCS 1B02G071800	thylakoid membrane protein TERC, chloroplastic	377	0	84.39	9.49	8.95	9.08	1.70	3.6E−18	2.67E−40	
TRAESCS 1B02G056700	translation initiation factor IF-2 isoform X1	239	7E−174	78.99	6.41	8.58	5.77	2.58	5.67E−56	9.72E−34	
TRAESCS5D02G537600	aspartokinase 1, chloroplastic-like	596	0	68.69	8.62	4.45	9.11	1.84	1.63E−37	2.03E−35	DEGs
in red boxes
of cluster 4
in Fig. 4I	
TRAESCS1B02G138100	auxin-responsive protein IAA15	198	2E-143	81.71	3.64	1.88	3.57	2.42	1.56E−28	1.12E−26	
TRAESCS5A02G247200	glucose-6-phosphate isomerase 1, chloroplastic	614	0	93.86	3.08	1.57	2.09	−0.27	2.07E−28	2.68E−36	
TRAESCS4A02G002600	MADS-box transcription factor 47-like isoform X2	163	7E-116	92.93	3.95	0.66	2.60	3.05	0.000023	0.000247	
TRAESCS7D02G246600	probable histone H2A variant 3	139	3E-95	95.7	2.08	1.78	3.65	2.75	5.65E−47	1.47E−26	
TRAESCS6B02G466700	protein argonaute 1C-like isoform X2	1013	0	89.28	5.24	2.58	4.52	5.89	1.53E−64	8.82E−62	
TRAESCS6D02G293200	putative MADS-domain transcription factor	228	1E-167	96.69	5.72	2.42	6.37	2.55	1.46E−42	1.31E−43	
TRAESCS1B02G100800	RNA-binding protein 8A	209	9E-152	85.74	4.77	1.01	4.64	2.08	4.73E−34	2.42E−40	
TRAESCS1B02G105100	ribosome biogenesis protein NOP53	407	0	83.3	4.30	1.07	6.42	2.08	2.01E−38	4.76E−32	
TRAESCS1B02G051600	uncharacterized protein LOC109787361	466	0	62.52	6.65	9.45	5.94	3.47	6.62E−46	1.2E−44	
Notes.

Bold numbers indicate more than two-fold changes in expression.

Figure 5 Gene set enrichment analysis with Kyoto Encyclopedia of Genes and Genomes (KEGG) pathways of the differentially expressed genes (DEGs).

Dots represent significant KEGG pathways from the pairwise gene set enrichment analysis comparisons at each time point after exposure to salt stress. The size of the dots indicates the number of differential genes, while the color of the dots represents the p-values of enrichment analysis. The rich factor refers to the ratio of the number of DEGs in the pathway to the total number of genes. The size of the dots indicates the number of genes, and the color of the dots indicates the −log10 FDR value within the pathway.

In addition to GO and KEGG analysis, the role of DEGs as transcription factors was investigated. DEGs at different timepoints under salt stress in PL1 and PL6 were identified using PlantTFDB (http://planttfdb//planttfdb.gao-lab.org). In total, 255 genes were identified with an e-value threshold of 1 × 10−1 and a sequence identity of >80% and further selected to compare the expression patterns between PL1 and PL6 under salt stress conditions. The most abundant type of transcription factor was the ethylene-response factor (ERF) protein family, followed by the basic helix-loop-helix (bHLH) protein family; heat shock transcription factor protein family; myeloblastosis (MYB)-related protein family; and Nam, ATAF, and CUC (NAC) protein family (Fig. 6A). Moreover, 255 putative transcription factors were grouped by expression pattern into six clusters and an unclassified group (Fig. 6B). Overall, 72, 44, and 35 DEGs were annotated by the ERF, bHLH, and MYB (related) protein families, respectively. These three transcription factors accounted for 59% of the total number of transcription factors. The expression patterns of DEGs in clusters 2 and 6 (marked with red boxes in Fig. 6B) were selected and expressed in heatmaps (Fig. 6C) to display differences in the expression patterns of DEGs between PL1 and PL6 under salt stress conditions (Table S7). Notably, PL6 exhibited higher expression of specific transcription factors under salt stress conditions than PL1, as displayed in the heatmap (Fig. 6C).

Figure 6 Differentially expressed transcription factors (TFs) under salt stress treatment in PL1 and PL6.

(A) Distribution of TF family members among the differentially expressed genes (DEGs). The bar graph illustrates the number of TFs belonging to each TF family among the DEGs. (B) Expression patterns of TFs at each time point after exposure to salt stress. Each cluster with similar expression patterns is indicated by red boxes. (C) Heatmap analysis of TF family genes in PL1 and PL6 under salt stress treatment, with the genes marked by red boxes in (B) specifically highlighted.

Lastly, 22 protein kinase genes were identified with significant expression patterns at different timepoints, including two calcineurin B-like (CBL)-interacting protein kinases and one mitogen-activated protein kinase (MAPK) with more than two-fold changes in PL6 under salt stress (Table 2). Additionally, 70 differentially expressed salt stress-responsive genes involved in regulating the circadian clock system, cytoskeleton organization, and cell wall organization were identified using MapMan, with 15 of them showing more than a two-fold change in PL6 (Table 3).

Table 2 List of differentially expressed protein kinase genes under salt stress condition.

Gene ID	Description	Length	E-value	Similarity (%)	Log2 fold change (PL6/PL1)	p-value	FDR	
					0 h	3 h	12 h	24 h			
TraesCS1B02G098700	CBL-interacting protein kinase 17	466	0	89.58	5.64	1.70	4.46	1.42	3.76E−43	6.32E−41	
TraesCS5B02G223900	CBL-interacting protein kinase 16	447	0	88.69	0.66	0.32	−0.20	−0.80	0.00187	0.011796	
TraesCS4B02G319900	CBL-interacting protein kinase 9	443	0	94.32	1.37	0.71	0.36	0.35	4.87E−26	3.05E−24	
TraesCS1B02G098600	CBL-interacting protein kinase 17	466	0	89.63	5.45	1.75	6.91	0.84	2.73E−25	1.65E−23	
TraesCS5A02G492000	CBL-interacting protein kinase 9	446	0	94.25	0.72	0.47	0.04	0.28	8.66E−13	2.34E−11	
TraesCS1D02G082500	CBL-interacting protein kinase 17	480	0	87.23	−0.21	0.29	−0.62	−0.89	3.29E−05	0.000341	
TraesCS4D02G118500	CBL-interacting protein kinase 14	362	0	82.78	0.67	1.11	0.31	0.33	0.00488	0.026029	
TraesCS1D02G082600	CBL-interacting protein kinase 17	448	0	86.72	0.36	0.01	−0.50	−0.37	6.13E−05	0.000598	
TraesCS1A02G080600	CBL-interacting protein kinase 17	466	0	90.22	−0.39	−0.16	−0.76	−1.21	1.96E−07	0.000003	
TraesCS1A02G080700	CBL-interacting protein kinase 17	471	0	89.6	−0.26	0.23	−0.86	−0.50	1.12E−10	2.52E−09	
TraesCS4B02G120400	CBL-interacting protein kinase 14	444	0	92.95	−0.12	−0.80	0.86	0.38	0.000271	0.002245	
TraesCS2D02G107100	CBL-interacting protein kinase 29	436	0	87.67	−0.24	−0.31	−0.51	0.08	0.005552	0.02886	
TraesCS3B02G169300	CBL-interacting protein kinase 5	464	0	93.17	0.61	0.96	0.16	−0.54	4.65E−08	7.74E−07	
TraesCS4D02G316500	CBL-interacting protein kinase 9	445	0	94.22	1.03	0.96	−0.10	0.27	3.23E−12	8.38E−11	
TraesCS3D02G151500	CBL-interacting protein kinase 5	464	0	93.02	0.59	0.71	0.13	−0.26	2.39E−05	0.000254	
TraesCS3A02G135500	CBL-interacting protein kinase 5	466	0	92.32	1.13	0.05	0.14	0.00	4.62E−06	5.69E−05	
TraesCS1B02G104900	mitogen-activated protein kinase 14	549	0	92.96	3.98	1.64	3.09	1.98	6.05E−42	9.68E−40	
TraesCS7A02G410700	mitogen-activated protein kinase 12	578	0	92.91	0.41	0.32	0.07	0.55	7.32E−06	8.69E−05	
TraesCS5B02G075800	SNF1-type serine-threonine protein kinase	363	0	93.99	0.85	0.85	0.20	0.92	5.49E−05	0.000542	
TraesCS5D02G081700	SNF1-type serine-threonine protein kinase	364	0	94.37	0.54	0.77	0.22	0.52	0.000294	0.002407	
TraesCS1D02G308200	SNF1-related protein kinase regulatory subunit beta-1	280	0	82.88	−1.25	−0.65	0.17	0.67	7.37E−05	0.000706	
TraesCS5A02G069500	SNF1-type serine-threonine protein kinase	360	0	95.18	−0.78	−0.29	−0.35	0.67	0.002772	0.01638	
Notes.

Bold numbers indicate more than two-fold changes in expression.

Table 3 List of differentially expressed salt stress responsive ggenes selected by the MapMan program.

Gene ID	Description	Length	E-value	Similarity (%)	Log2 fold change (PL6/PL1)	p-value	FDR	BinName from MapMan	
					0 h	3 h	12 h	24 h				
TraesCSU02G196100	Pseudo-response regulator (PRR)	660	0	97.11	−0.67	−0.01	−0.99	−4.44	4.11E−07	5.98E−06	Circadian clock system	
TraesCS5D02G078500	Kinesin-like protein KIN-12F isoform X2	3015	0	84.6	−0.73	−0.21	−0.05	−2.42	7.62E−15	2.4E−13	Cytoskeleton organization	
TraesCS1B02G123200	Kinesin-like protein KIN-13A	519	0	92.31	5.27	2.00	5.67	2.61	1.73E−62	8.72E−60	Cytoskeleton organization	
TraesCS1B02G024500	Actin-7	377	0	99.58	4.70	4.81	5.63	2.86	4.55E−63	2.37E−60	Cytoskeleton organization	
TraesCS5B02G491800	Actin depolymerization factor-like protein	147	6.3E−104	88.78	0.86	−0.12	0.73	−2.31	1.92E−05	0.00021	Cytoskeleton organization	
TraesCS5D02G492300	Actin depolymerization factor-like protein	147	6.3E−105	87.63	0.55	0.09	0.38	−3.25	2.08E−11	5.07E−10	Cytoskeleton organization	
TraesCS1B02G069300	Protein unc-13 homolog	1107	0	93.26	9.87	9.73	10.07	1.73	1.97E−53	5.97E−51	Cell wall organization	
TraesCS6D02G048900	Melibiase family protein	637	0	80.94	−2.04	−3.80	−2.48	−0.04	0.000177	0.001532	Cell wall organization	
TraesCS1B02G084600	Hydroxyproline O-galactosyltransferase GALT3	591	0	83.91	3.76	2.71	2.78	0.79	5.51E−16	1.89E−14	Cell wall organization	
TraesCS1D02G019000	Tricin synthase 1	248	1.8E−180	79.05	−1.48	0.37	−2.29	−0.17	6.39E−09	1.18E−07	Cell wall organization	
TraesCS5D02G488900	Caffeic acid O-methyltransferase	353	0	86.01	−1.86	−2.56	−4.53	0.63	7.23E−20	3.17E−18	Cell wall organization	
TraesCS1B02G098800	Acyl transferase 4	435	0	80.27	8.49	2.00	8.69	2.57	4.28E−31	3.63E−29	Cell wall organization	
TraesCS3D02G116600	Alkane hydroxylase MAH1-like	517	0	88.56	0.02	0.08	−2.60	−2.99	0.001075	0.00741	Cell wall organization	
TraesCS5D02G127300	Aldehyde dehydrogenase family 3 member H1-like	479	0	86.99	0.49	−0.29	−2.17	−1.78	2.63E−10	5.72E−09	Cell wall organization	
TraesCS2A02G045800	GDSL esterase/lipase LTL1	369	0	90.05	9.20	1.62	8.58	8.50	3.69E−36	4.22E−34	Cell wall organization	
Notes.

Bold numbers indicate more than two-fold changes in expression.

Enzyme activities assays

To investigate the differences in enzyme activities between PL1 and PL6 under salt stress conditions, we measured CAT, POD, and SOD activities and TAC (Fig. 7). Upon subjecting both wheat lines to salt stress, we observed distinct patterns in enzyme activities. In PL6, CAT and POD activities significantly increased after 24 and 48 h of exposure to salt stress (Fig. 7A and 7B). Conversely, in PL1, SOD activity slightly decreased after 24 and 48 h exposure to salt stress (Fig. 7C). Furthermore, the TAC in PL1 was not significantly changed by salt stress (Fig. 7D). Conversely, in PL6, the TAC notably increased after 24 and 48 h of exposure to salt stress. This increase in TAC suggests that PL6 has a higher capacity to counteract oxidative stress and maintain cellular redox balance than PL1, contributing to its enhanced salinity tolerance.

Figure 7 Biochemical assays of antioxidant enzyme activity.

(A) Catalase (CAT) activity, (B) peroxidase (POD) activity, (C) total superoxide dismutase (SOD) activity, and (D) total antioxidant capacity (TAC). Each bar represents the average ±standard error (n = 3). Independent t-tests demonstrated significant differences (*p < 0.05 and **p < 0.01) compared to the control condition (0h).

Validation of the DEG results using reverse transcription-quantitative polymerase chain reaction

Supporting the DEG results, 12 genes from the three aforementioned clusters from PL1 and PL6 were selected for RT-qPCR (Fig. 8). All the selected genes were more highly expressed in PL6 than in PL1. peroxidase 2 (TRAESCS1B02G095800), nitrate transporter (TRAESCS1B02G038700), auxin-responsive protein (TRAESCS1B02G138100), and replication protein A (TRAESCS1B02G102200) transcripts in PL6 were highly expressed at 48 h following salt treatment (Fig. 8A). Nuclear transport factor 2- like protein (TRAESCS2A02G046200), histone H2A (TRAESCS1B02G048900), integral membrane protein (TRAESCS1B02G071800), and histone H2A variant 3 (TRAESCS7D02G246600) transcripts in PL6 continuously decreased at 24 h and peaked at 48 h following salt treatment (Fig. 8B). Argonaute 1C-like isoform X2 (TRAESCS6B02G466700), MADS-box (TRAESCS6B02G017900), and aspartokinase 1 (TRAESCS5D02G537600) transcripts in PL6 peaked at 3 h, and all gradually decreased, except for ribosome biogenesis protein NOP53 (TRAESCS1B02G105100) (Fig. 8C). These results are consistent with those of RNA sequencing (RNA-seq).

Figure 8 Validation of the RNA sequencing results via reverse transcription-quantitative polymerase chain reaction (RT-qPCR) at different timepoints under salt stress conditions.

Three clusters representing different expression patterns for PL1 and PL6 were selected and the relative expressions shown. RT-qPCR was performed with three biological replicates. Each bar represents the average ± standard error (n = 3). Independent t-tests showed significant differences (*p < 0.05 and **p < 0.01).

Discussion

This study revealed that salinity stress had negative effects on germination and plant growth during the developmental process. Na+ is considered a nonessential element in plants (Nieves-Cordones, Al Shiblawi & Sentenac, 2016); however, excessive accumulation of Na+ can have detrimental effects on plants, including disrupting cellular homeostasis, inducing oxidative stress, and suppressing growth (Munns & Tester, 2008; Craig Plett & Møller, 2010). In this study, PL6 consistently showed higher germination rates and better seedling growth under salt stress than PL1. The differences in K+ and Na+ contents between PL1 and PL6 supported these observations, highlighting PL6′s superior performance under salt treatment conditions. Previous studies have emphasized the significance of maintaining a low sodium concentration and a high K+/Na+ ratio as crucial traits in plant salt tolerance (Ismail & Horie, 2017; Naeem et al., 2020). Salinity triggers osmotic stress, resulting in accumulation of higher salt levels in cell sap and tissues (EL Sabagh et al., 2021). This accumulation is directly linked to the buildup of Na+ and Cl−, causing ion imbalance that adversely affects germination and subsequent metabolic processes (Hussain et al., 2019). Genome-wide transcriptomic analysis has emerged as a powerful tool to investigate stress-tolerant genes, gene families, and related mechanisms in plants (Peng et al., 2014; Li et al., 2016). In this study, we observed a significant difference in the salinity response between PL1 and PL6 and identified distinct expression patterns of DEGs between the two lines. Although a higher number of DEGs was found in PL6 compared with PL1, it is important to note that the majority of these DEGs exhibited similar expression patterns in both PL1 and PL6 under salt stress conditions. This could be because PL6 was generated through a mutation of PL1 via gamma irradiation, leading to the sharing of numerous genomes between them. Nonetheless, despite the similar expression patterns, clear phenotypic differences were observed, including variations in germination rate, shoot and root growth, and ion concentrations (Na+ and K+). Thus, our genome-wide transcriptional analysis allowed us to identify the DEGs responsible for the differential responses of PL1 and PL6 under salt stress conditions.

Salt stress not only induces osmotic stress but also leads to ionic imbalance, resulting in ion toxicity and, ultimately, the production of ROS (Julkowska & Testerink, 2015). In our study, PL1 (as the wild-type line) exhibited a dark-purple seed coat and had high levels of anthocyanin, phenolic compounds, and antioxidant activities (Hong et al., 2019). Similarly, PL6, which was generated by irradiating PL1 with 200 Gy of gamma rays, also displayed a dark-purple seed coat.

As shown in Fig. 3D, GSEA revealed several GO terms that were positively correlated with salt stress, including defense response (GO: 0006952), glutathione metabolic process (GO: 0006749), peroxidase activity (GO: 0004601), ROS metabolic process (GO: 0072593), response to biotic stimulus (GO: 0009607), and response to stress (GO:0006950). Among these terms, three were specifically related to antioxidant activity: glutathione metabolic process (GO: 0006749), peroxidase activity (GO: 0004601), and ROS metabolic process (GO: 0072593). These antioxidant-related GO terms are crucial protective mechanisms against salinity stress in plants. Interestingly, we observed that DEGs related to antioxidants were specifically upregulated in PL6 48 h after salt stress, despite both PL1 and PL6 having colored seed coats. This suggests that these DEGs may positively contribute to salt stress tolerance, leading to more vigorous shoot and root growth in PL6 than that in PL1. In addition to the gene expression analysis, the measurement of antioxidant enzyme activities further supports the higher antioxidant capacity in PL6 than in PL1 under salt stress conditions. CAT and POD activities were significantly increased at 24 and 48 h after salt stress exposure in PL6 (Fig. 7A and 7B), indicating efficient ROS-scavenging ability and peroxide detoxification, which help protect the cells from oxidative damage during salt stress. Conversely, in PL1, SOD activity slightly decreased at 24 and 48 h post-salt stress (Fig. 7C), suggesting a limited ability to efficiently neutralize superoxide radicals, potentially leading to ROS accumulation and oxidative stress in PL1 under salt stress conditions. Overall, these findings not only provide insights into the DEGs related to antioxidant activity but also highlight the distinctive enzymatic responses to salt stress in PL1 and PL6. The increases in CAT and POD activities and TAC in PL6 might play crucial roles in its superior ability to manage salt-induced oxidative stress compared with the wild-type PL1. The combination of gene expression analysis and antioxidant enzyme activity measurements sheds light on the activation of specific antioxidant pathways in PL6, providing a comprehensive understanding of its enhanced salinity stress response.

Phytohormones, such as abscisic acid (ABA) and auxins (indole acetic acid [IAA] and indole-3-butyric acid), play crucial roles in plant responses to environmental stresses, including salinity. ABA promotes ABA-dependent and reactive oxygen species ROS-related stress signaling mechanisms, conferring salinity tolerance in wheat (Dong et al., 2013). IAA play an important role in the metabolism of plants by increasing the activity of enzymes responsible for biosynthesis of growth promoters under saline stress in wheat (Hendawey, 2015). In this study, we observed increased transcription levels of TRAESCS1B02G145800 (ABA receptor PYL8) and TRAESCS1B02G138100 (auxin-responsive protein IAA15) in PL6 under salt stress conditions (Table 1). Additionally, salinity-induced osmotic stress leads to the overproduction of ROS and oxidative damage to plant cells. To counteract this, the antioxidant defense system in plants is activated to detoxify ROS and maintain redox homeostasis (Hasanuzzaman et al., 2021). Accordingly, we found that plant hormone-related genes, including dehydroascorbate reductase and peroxidase genes, were upregulated in PL6 under salt stress to protect against ROS-induced damage and maintain cellular redox balance (Table 1). The increased expression of ROS-related genes in PL6 suggests that this mutant line may exhibit an altered response to salt stress-induced oxidative stress.

In addition to hormone-related responses, transcriptional regulation through histone modification and chromatin remodeling plays a pivotal role in plant responses to salt stress. In this study, we observed an increase in the transcription levels of INO80 complex subunit D (TRAESCS1B02G149000) in PL6 under salt stress conditions. The INO80 chromatin remodeling complex is responsible for evicting the histone variant H2A.Z in eukaryotic cells (Alatwi & Downs, 2015). Studies in Arabidopsis have demonstrated that under salt stress, the INO80 complex induces the eviction of H2A.Z-containing nucleosomes from the AtMYB44 promoter region, leading to increased accumulation of AtMYB44 transcripts and thus promoting salt stress tolerance (Nguyen & Cheong, 2018). However, the specific target gene and position of the histone variant H2A.Z evicted by the INO80 complex in wheat remain unclear. Further investigations are required to identify the precise position of H2A.Z evicted by the INO80 complex and clarify the factors influencing the differential responses of PL1 and PL6 to salinity stress.

Moreover, investigation of the MADS-box family members contributes to our understanding of the differential responses of PL1 and PL6 to salinity stress. MADS-box transcription factors are known to regulate flowering development (Lee & Lee, 2010; Callens et al., 2018). Wu et al. (2020) reported that overexpression of OsMADS25 in rice and Arabidopsis resulted in improved salinity tolerance compared with that in the wild-type. Conversely, the MADS-box transcription factor AGL16 was identified as a negative regulator of stress responses in Arabidopsis (Zhao et al., 2021). In this study, we observed increased transcription levels of two MADS-box transcription factors (TRAESCS4A02G002600 and TRAESCS6D02G293200) in PL6 mutant plants under salt stress conditions, suggesting their potential roles in salt tolerance and growth response. These findings provide valuable insights into the molecular mechanisms underlying the differential responses of PL1 and PL6 to salinity.

Furthermore, although GO terms related to photosynthesis were detected via GSEA and network analysis (Figs. 3D and 4B), no significant differences were observed in the gene expression patterns between PL1 and PL6. This finding is consistent with the data on chlorophyll concentration (Fig. 2E), which did not show significant variation between PL1 and PL6 during the duration of salt stress exposure. In our previous study, we observed that the total anthocyanin concentrations in wheat mutant lines (used in this study) were significantly higher than those in wild-type lines, resulting in higher antioxidant activity in the mutants compared with the wild-type (Hong et al., 2019). In the present study, the enriched factors “Biosynthesis of secondary metabolites” and “Flavonoid biosynthesis” increased following salt stress treatment (Fig. 5). This suggests that the antioxidant activities of PL6 under salt stress conditions might be influenced by these pathways, which include genes associated with GO terms such as glutathione metabolic process (GO: 0006749), peroxidase activity (GO: 0004601), and ROS metabolic process (GO: 0072593) (Fig. 3D).

As shown in Fig. 4D, several DEGs were mapped to GO terms related to gene expression regulation (GO: 001046), DNA binding transcription factor activity (GO: 0003700), and transcription regulator activity (GO: 0140110). To elucidate the molecular mechanism of salt stress response at the cellular level, we analyzed putative transcription factors and selected those with differential expression patterns in PL6 under salt stress conditions. Among them, the ERF family protein emerged as an important family of transcription factors in plants, regulating various developmental processes (Nakano et al., 2006), including their response to salt stress (Cheng et al., 2013; Li et al., 2020b; Trujillo et al., 2008). Additionally, studies have revealed the significance of the bHLH and MYB gene families in the response to salt stress in plants (Yang et al., 2021; Li et al., 2020a; Jiang, Yang & Deyholos, 2009; Kim et al., 2013; Seo et al., 2012). The putative transcription factors shown in Fig. 6 can be further analyzed for their functions to better understand the molecular mechanisms of salt response. Flavonoid biosynthesis has been extensively studied and is predominantly regulated at the transcriptional level by the MYB–bHLH-WD40 complex in various plant species, such as rice, Arabidopsis, Mimulus, apples, and maize (Tohge, de Souza & Fernie, 2017; An et al., 2020; Yuan et al., 2014; Zheng et al., 2019; Baudry, Caboche & Lepiniec, 2006). In this study, several bHLH and MYB gene families were identified as putative transcription factors, likely influenced by the seed colors of PL1 (wild-type) and PL6 (mutant line) used in the experiment. Consequently, based on the heat map in Fig. 6, the bHLH and MYB gene families exhibiting different expression patterns between PL1 and PL6 were considered differentially expressed transcription factors under salt stress conditions.

Moreover, protein kinases play a vital role in regulating plant responses to salt stress. Jin et al. (2016) investigated the expression levels of protein kinase genes in response to salt stress in transgenic wheat. They found that overexpression of TaCIPK25 resulted in hypersensitivity to Na+ and superfluous accumulation of Na+ in transgenic wheat lines. Similarly, the overexpressed TMPK3 (Triticum aestivem Mitogen Activated Protein Kinase) promotes salt and osmotic stress tolerance to levels exceeding those observed in wild type plants. This tolerance is associated to a lower sensitivity to exogenous ABA, and increased stronger accumulation of proline contents, higher survival, and lower water loss rates as well as attenuated oxidative stress status (Ghorbel et al., 2023). These findings underscore the importance of protein kinases in the regulation of plant responses to salt stress and suggest that different types of protein kinases play specific roles in these processes.

In the present study, we identified 15 DEGs with more than a two-fold change, among which one, five, and nine genes were involved in the circadian clock system, cytoskeleton organization, and cell wall organization, respectively (Table 3). These processes play crucial roles in plant stress response and are important components of how plants adapt to challenging environments. The circadian clock system has been found to be essential in regulating the plant’s response to salt stress. Xu, Yuan & Xie (2022) conducted a study on Arabidopsis plants and demonstrated that the circadian clock system is involved in the modulation of salt stress responses. They observed altered expression levels of circadian clock genes under salt stress conditions and further noted that the disruption of the circadian clock system resulted in reduced salt tolerance in the plants. Likewise, the cytoskeleton organization is also critical for regulating plant responses to salt stress. For instance, in rice plants, the actin cytoskeleton has been shown to play a role in regulating the response to salt stress, ion homeostasis, and ROS scavenging (Chun et al., 2021). Disruption of the actin filaments in rice plants led to reduced salt tolerance, indicating the importance of the cytoskeleton in coping with salt-induced stress. Moreover, the cell wall organization is a vital aspect of the response of maize to salt stress. A study on maize revealed that the expression of genes related to the cell wall was altered under salt stress conditions, and modification of the cell wall composition contributed to increased salt tolerance in the plants (Oliveira et al., 2020). These findings highlight the significance of the cell wall in mediating the plant’s ability to withstand salt stress. Building upon this understanding, further research incorporating histological analyses of root tissues and transcriptomic profiling of roots would be instrumental in unraveling the genetic basis and tissue-level adaptations responsible for the superior salt stress response and tolerance of PL6. This integrated approach could provide valuable insights into the intricate mechanisms governing salt tolerance in plants and contribute to the development of more resilient crop varieties.

Conclusions

Consequently, in this study, the salt-tolerant colored wheat mutant PL6 was selected, and its salt resistance mechanism was scrutinized through transcriptome analysis, comparing PL6 with wild-type wheat (PL1). The findings of this study provide valuable insights into salt tolerance breeding in wheat, offering diverse interpretations of salinity. This comprehensive investigation not only enhances our understanding of the associated molecular mechanisms but also carries practical implications for the development of crops under saline conditions. Further research incorporating histological analyzes of root tissues and transcriptomic profiling of “roots would be instrumental in unraveling the genetic basis and tissue-level adaptations responsible for the superior salt stress response and tolerance of PL6.

Supplemental Information

Supplemental Information 1 Supplementary Figure 1 (Fig. S1). Field images of M6 generations of PL1 and PL6 at different time points

Supplemental Information 2 Comparison of seed coat color of different wheat lines

PL1 (control) and PL6 (mutant line) were used in this study.

Supplemental Information 3 Comparison of salt stress response in mutant lines (PL2-PL49) and wild type control (PL1)

(A) Germination rate of mutant lines (PL2-PL49) and PL1 as the wild type control. (B) Shoot length of mutant lines (PL2-PL49) and PL1 as the wild type control. (C) Root length of mutant lines (PL2-PL49) and PL1 as the wild type control. For the preliminary screening of the selected mutant lines, 100 seeds from each line were placed in a phytohealth chamber (SPL Life Sciences) with two layers of germination paper, and a total volume of 200 ml of the solution containing 150 mM NaCl was applied to them at a temperature of 22° C. After 4 days, the germination rate, shoot length, and root length were recorded. PL1 served as the wild type control in these experiments.

Supplemental Information 4 The details of the primers used in this study

Supplemental Information 5 Germination ratio of the different salt concentrations on seed germination

Supplemental Information 6 Summary of RNA-seq quality, read counts, and mapping rates

Supplemental Information 7 Differentially expressed genes (DEGs) from BlastX results against NCBI Poaceae family

This table contains the blastx results against the NCBI Poaceae family, which led to the identification of a total of 4,017 differentially expressed genes (DEGs) with a p-value ¡ 0.05 and FDR ¡ 0.05.

Supplemental Information 8 Gene Set Enrichment Analysis (GSEA) for PL1 and PL6 using gene ontology (GO) mapping of differentially expressed genes

Supplemental Information 9 Gene Set Enrichment Analysis (GSEA) for PL1 and PL6 using Kyoto Encyclopedia of Genes and Genomes (KEGG) pathways mapping of differentially expressed genes

Supplemental Information 10 Differentially expressed genes in red boxes of Fig.6B used in Fig. 6C

Supplemental Information 11 Raw Data for Figures 1, 2, 7, and 8

Supplemental Information 12 Essential MIQE Checklist

Additional Information and Declarations

Competing Interests

Author Contributions

Data Availability

The authors declare there are no competing interests.

Min Jeong Hong conceived and designed the experiments, performed the experiments, analyzed the data, prepared figures and/or tables, authored or reviewed drafts of the article, and approved the final draft.

Chan Seop Ko performed the experiments, prepared figures and/or tables, and approved the final draft.

Jin-Baek Kim analyzed the data, authored or reviewed drafts of the article, and approved the final draft.

Dae Yeon Kim conceived and designed the experiments, analyzed the data, prepared figures and/or tables, authored or reviewed drafts of the article, and approved the final draft.

The following information was supplied regarding data availability:

The FASTQ files of raw data is available at NCBI SRA: PRJNA937396.

The raw data are available in the Supplemental File.

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
