# Peer review of "Identification and transcriptomic profiling of salinity stress response genes in colored wheat mutant"

_PeerJ, doi:10.7717/peerj.17043_

## Round 0.1 · original submission · Minor Revisions

Dear authors,

Your manuscript touches on an important issue. Well constructed and written. But it needs a few corrects, which reviewers mentioned.

Additionally, here are the points I would like you to add:

Line 98: Specify which species you mean by "hexaploid wheat". Write the scientific name. Is it different from Triticum aestivum?

Line 576-579: Add emphasis on salt stress tolerance in the last lines of the conclusion.

·

Basic reporting

In the introduction section, the first paragraph should be replaced with the second paragraph. In the abstract section, the word "stress" is in Line 15. Salinity stress expression should be removed. The aim of this study is not clear.
Although genetic diversity analyses have not been performed, the importance of this is mentioned in the introduction section, and the introduction section should be written as much as possible for the study performed. In fact, the introduction part is halfway well-constructed. Of course, if the first paragraph is replaced by the second paragraph. The introduction part needs to be revised again and irrelevant sections need to be removed

Experimental design

In line 98, instead of hexaploid wheat is better to use “wheat is a hexaploid (2n=6x=42)”, what is M6 generation? Do you mean the RIL population?
Line 114-118. (Among the tested mutants, one specific mutant, named PL6, demonstrated exceptional salt tolerance, exhibiting a high germination rate and favorable growth characteristics. Therefore, PL6 was chosen for further detailed analysis in the context of salt tolerance, and the hexaploid wheat inbred line PL1 was used as the control line. Those phrases should be in the results section, they are results, neither material nor methods.
Please cite this phase “Hoagland’s culture solution).
The total volume of 200 mL of the solution containing 150 mM NaCl, such expression should be in the abstract also.

Validity of the findings

The discussion section should be well-supported by the literature, providing a thorough and insightful analysis. It is crucial to discuss differences and similarities based on existing literature. Assumptions about the reasons behind the differences should be made, and each distinct aspect of the article should be supported by relevant literature. I
t is important to remember that similarities merely indicate that the research is on the right track, whereas differences prompt the researcher to think critically.
The discussion section is extensive and well-crafted, but it should be adequately substantiated with literature.

Half of the first paragraph of the results section technically belongs to the material and methods section.

Line 254-255. To evaluate the changes in Na+ and K+ ion contents in response to salt stress, wheat leaves were collected at 3, 24, and 48 h after salt treatment. This not results, is a method sentence.

Also “To identify the differences in gene ontology (GO) term enrichment between PL1 and PL6 during salt stress, all the DEGs of PL1 and PL6 at different time points were analyzed using GSEA with a default parameter. This not results, is a method sentence.

Line 299-303 lines. Give the table in brackets, please!
Line 703-704. The journal should be in italics
Line 783-786. The journal should be in italics
Line 837-839. The journal should be in italics
Please have a check to such rules problems.

Reviewer 2 ·

Basic reporting

Corrections and suggestions are given in the text. No comment

Experimental design

This is original research within Aims and Scope of the journal. It is a meticulous work performed to a high technical and ethical standard. It includes methods sufficient detail and information. Some corrections and suggestions are given in the text.

Validity of the findings

Suitable and sufficient. Some corrections and suggestions are given in the text.

Additional comments

Some corrections and suggestions are given in the text.

Annotated reviews are not available for download in order to protect the identity of reviewers who chose to remain anonymous.

Reviewer 3 ·

Basic reporting

The article was written in English, but the language and some spellings must be improved—for example, the spelling of e-value at line 317. A few other such errors are scattered throughout the manuscript.

The authors relied on several old references, some more than 30 years ago, while recent references exist. I think new and recent references should be used. In some instances, very categorical statements were made without citation. For example, in line 41 the authors stated...."The impact of salinity is extensive, with >20% of irrigated land worldwide being affected" This statement, I believe, should be attributed to an authority.

Experimental design

There seem to be some omissions in some methodologies. For example, in line 129, it is stated that the leave samples were collected at 3, 24, and 48 hours. But in line 156, it was said that the leave samples were collected at 0, 3, 24, and 48 hours.

Similar issues can be found in other parts of the manuscript.

Validity of the findings

From line 275-280, the number of DEGs identified were stated. It is however not very clear to me the exact time point at which those genes were differentially expressed.

---

## Round 0.2 · Minor Revisions

There are still shortcomings in your corrections.

In the abstract section, delete the background, results and conclusion titles.

In the Introduction section, you changed the position of the paragraphs according to the reviewer's comments, but the fluency was disrupted and away from logical. The reviewer may have intended to remove this paragraph. In my opinion, the best way is to write it as follows.

Wheat is a crucial crop cultivated globally, contributing to 30% of global grain production and providing approximately 20% of the calories consumed by humans (Shiferaw et al., 2013; Seleiman et al., 2022). Soil salinity poses a critical issue, resulting in yield losses of up to 60% in wheat production (El-Hendawy et al., 2017). The impact of salinity is extensive, with >20% of irrigated land worldwide being affected (EL Sabagh et al., 2021). Furthermore, it is projected that up to 50% of arable land will be lost by 2050 owing to salinization caused by both human activities and ongoing climate change (Asif et al., 2018; Kumar & Sharma, 2020; Chele et al., 2021).

Salinity stress disrupts plant growth by increasing Na+ ion assimilation and reducing the Na+/K+ ratio, leading to osmotic stress and ion toxicity, consequently affecting normal plant development (EL Sabagh et al., 2021). Additionally, under salinity stress, oxidative stress can impair plant growth through reduced photosynthetic capacity, oxidative damage caused by an imbalance in reactive oxygen species (ROS) production, and decreased antioxidant activity, ultimately leading to reduced crop yield (Hasanuzzaman et al., 2014; Sadak, 2019; Omrani et al., 2022).

Line 57: I think you mean biotechnology not biology. Please, correct it.
Give literature at the end of sentences. Do not break sentences.
Delete lines 88-95. These sentences are not the purpose of your study. These are the results of your work. Write down your purpose.
For example; You should write a sentence like "The purpose of this study is to investigate the change of the salt tolerance mechanism in the wheat line we developed through gamma ray mutation."

Did the colored wheat (PL1) you used in your study emerge only as a result of hybridization? Are the parents also colored wheat? Do you have any data showing that colored wheat has high salt tolerance? Why did you choose colored wheat? Add information about it.

Line 104: "diversity" not "diversify"

Line 230: You did not use all wheat genotypes shown in Fig S2 in your study. Why did you include their photo? I find it irrelevant to work. In my opinion, it is enough to give the mutant line just to show the change with the application of mutation.

Line 233-235: Mention that you did a preliminary study in the material method section. You give the results here too.

Line 360-365: It is irrelevant that you give examples from different species of plants and talk about the differences between genotypes. If there are results previously presented on a genotype changed by mutation, find them and add them. Otherwise, why did you provide these when there are dozens of studies on wheat genotypes? The reviewers in the previous revision also asked you to support it with more current literature. Please support a literature on wheat and in recent years.

Line 407-413: See explanation above. Support with wheat-related and current literature. No more than 10 years old, please.

Line 477-490: See explanation above. Support with wheat-related and current literature. No more than 10 years old, please

Line 520-545: You can write these as the conclusion section. Write "in conclusion" instead of "in summary". Delete the conclusion completely. Too much repetitive information bores the reader. Simplify.

---

## Round 0.3 · Minor Revisions

Line 82-96: You did not write your purpose here. You have presented some introductory sentence, some method information, and some findings. It is not appropriate to write down your findings or make recommendations in the introduction section. Delete them completely.

Line 82: Start a new paragraph after the sentence "..................to develop salt-resistant wheat by securing genetic diversity."

In the paragraph you just started, "Colored wheat possesses advantages such as enhanced nutritional content, increased antioxidant levels, and potential health benefits due to its rich phytochemical profile. Colored wheat possesses advantages such as enhanced nutritional content, increased antioxidant levels, and potential health benefits due to its rich phytochemical profile (Hong et al., 2019; Garg et al., 2022). Start with the sentence " and then add the following sentence.

"Investigating the salt tolerance in colored wheat, which is characterized by these valuable features, and revealing the distinctive features that contribute to its resistance to salinity are essential to obtain new genetic resources with high salt tolerance and nutrition. This study aims to investigate the changes in the salt tolerance mechanism in the colored wheat line we developed through gamma-ray mutation."
After these sentences, start the material and method section without any explanation.

Line 517-523: Delete these sentences. Avoid repetitions. Writing the same thing in different sentences will not make your manuscript better.

You haven't changed the conclusion section at all. You should make it a little shorter and give your clear result. I think the conclusion could be as follows. "Consequently, in this study, the salt-tolerant colored wheat mutant PL6 was selected, and its salt resistance mechanism was scrutinized through transcriptome analysis, comparing PL6 with wild-type wheat (PL1). The findings of this study provide valuable insights into salt tolerance breeding in wheat, offering diverse interpretations of salinity. This comprehensive investigation not only enhances our understanding of the associated molecular mechanisms but also carries practical implications for the development of crops under saline conditions. Further research incorporating histological analyzes of root tissues and transcriptomic profiling of "roots would be instrumental in unraveling the genetic basis and tissue-level adaptations responsible for the superior salt stress response and tolerance of PL6." These sentences can be the last sentences of the conclusion section of your manuscript.

---

## Round 0.4 · accepted · Accept

Congratulations. Your manuscript is now acceptably improved.